# Comparative Analysis of Environment-Responsive Alternative Splicing in the Inflorescences of Cultivated and Wild Tomato Species

**DOI:** 10.3390/ijms231911585

**Published:** 2022-09-30

**Authors:** Enbai Zhou, Guixiang Wang, Lin Weng, Meng Li, Han Xiao

**Affiliations:** 1University of Chinese Academy of Sciences, 19A Yuquan Rd, Beijing 100049, China; 2National Key Laboratory of Plant Molecular Genetics, CAS Center for Excellence in Molecular Plant Sciences, Institute of Plant Physiology and Ecology, Chinese Academy of Sciences, 300 Fenglin Rd., Shanghai 200032, China

**Keywords:** alternative splicing, transcriptome, phenotypic plasticity, inflorescence, domestication, tomato (*Solanum lycopersicum*)

## Abstract

Cultivated tomato (*Solanum lycopersicum*) is bred for fruit production in optimized environments, in contrast to harsh environments where their ancestral relatives thrive. The process of domestication and breeding has profound impacts on the phenotypic plasticity of plant development and the stress response. Notably, the alternative splicing (AS) of precursor message RNA (pre-mRNA), which is one of the major factors contributing to transcriptome complexity, is responsive to developmental cues and environmental change. To determine a possible association between AS events and phenotypic plasticity, we investigated environment-responsive AS events in the inflorescences of cultivated tomato and its ancestral relatives *S. pimpinellifolium*. Despite that similar AS frequencies were detected in the cultivated tomato variety Moneymaker and two *S. pimpinellifolium* accessions under the same growth conditions, 528 genes including splicing factors showed differential splicing in the inflorescences of plants grown in open fields and plastic greenhouses in the Moneymaker variety. In contrast, the two *S. pimpinellifolium* accessions, LA1589 and LA1781, had 298 and 268 genes showing differential splicing, respectively. Moreover, seven heat responsive genes showed opposite expression patterns in response to changing growth conditions between Moneymaker and its ancestral relatives. Accordingly, there were eight differentially expressed splice variants from genes involved in heat response in Moneymaker. Our results reveal distinctive features of AS events in the inflorescences between cultivated tomato and its ancestral relatives, and show that AS regulation in response to environmental changes is genotype dependent.

## 1. Introduction

Plants constantly cope with environmental perturbations during their life cycles. To survive in harsh environments, plants often display growth plasticity [1]. The physiological changes of plants in response to environmental cues are associated with altered gene expression, reflected by transcriptome dynamics [2,3,4,5]. For example, cultivation locations together with climate conditions have significant impacts on the transcriptomes of genetically identical grapevine (*Vitis vinifera*) clones, with the expression of 5% of the grapevine protein coding genes being environment sensitive [2]. Similarly, field type and macroclimate, especially solar radiation and temperature, affect transcriptome dynamics in rice [5].

The alternative splicing (AS) of precursor message RNA (pre-mRNA) is one of the major factors affecting transcriptome complexity and also has a profound impact on protein coding capacity [6,7,8,9]. In general, 40–70% of plant multi-exon genes have been predicted to undergo AS [10,11,12,13,14,15,16,17,18,19,20,21,22,23]. AS regulates not only developmental processes but also plant response to biotic and abiotic stresses [18,24,25,26,27,28,29,30,31,32,33]. Plant growth conditions such as light, temperature and nutrient supply greatly affect AS events. For example, when dark-grown Arabidopsis seedlings are subjected to light treatments, several hundred AS events were affected [34]. Furthermore, splicing factor serine/arginine-rich (SR) genes undergo AS and produce a considerable number of splice variants in response to stresses [35].

The tomato (*Solanum lycopersicum*) is an important vegetable crop grown worldwide. A considerable portion of tomatoes are produced in greenhouses with relatively stable temperature and humidity, contrasting to their ancestral relatives that thrive where environmental conditions change constantly and often dramatically and unpredictably. Flower number per inflorescence is a stable trait when plants are grown in well-controlled environments, but the trait also displays strong phenotypic plasticity in response to different environments [36,37]. We previously reported that there are large variations in the developmental plasticity of inflorescence between cultivated tomato and its ancestral relatives *S. pimpenillifolium* when grown under controlled conditions; cultivated tomatoes showed a small change in flower number per inflorescence, whereas *S. pimpenillifolium* accessions displayed strong inflorescence plasticity [36]. For example, the cultivar Moneymaker produces 7 flowers on each inflorescence when grown in a phytotron, whereas it produces 5–11 flowers under plastic greenhouse conditions. In contrast, the number of flowers per inflorescence in *S. pimpenillifolium* may increase from 15 to more than 30 and often the inflorescences become indeterminate. Flower number per inflorescence is determined by the number of flowers formed on each inflorescence branch and the number of branches and it has significant impact on fruit number—a major element in determining tomato yield. Several genes have been identified for their regulatory roles in inflorescence branching, including *SlWOX9/S*, *ANANTHA (AN)*, *FALSIFLORA* (*FA*), the *SEPALLATA*-like genes *JOINTLESS 2* (*J2*), *ENHANCE of J2 (EJ2),* the *SOC1*-like genes *TM3* and *STM3* and the *FUL*-like genes *MBP20, FUL1* and *FUL2* [38,39,40,41,42,43,44,45]. Of these genes, *SlWOX9/S*, *AN* and *FA* control inflorescence branching through the regulation of the timing of floral meristem maturation [39]. It remains unknown whether any of the abovementioned genes are involved in the regulation of the plasticity of inflorescence development in tomatoes.

Moreover, modern crop varieties have been bred for growing in optimized environments and have relatively stable phenotypes. To breed new varieties suitable for growing in suboptimal environments under the current trend of global climate change, it is important to understand how domestication and breeding has changed phenotypic plasticity. Notably, recent studies have revealed that gene expression regulated by AS plays crucial roles in plant response to changing environments, especially varying temperature [26,30,46,47,48,49,50]. In tomatoes, intronic polymorphisms affecting AS of the heat stress transcription factor *HsfA2* have been found to be associated with the difference in heat stress response between cultivated tomato and wild species [51]. To test whether AS is involved in the regulation of the plasticity of inflorescence development, we performed a comparative analysis of AS events in the inflorescences of the cultivated tomato Moneymaker and its two ancestral relatives *S. pimpenillifolium* accessions, LA1589 and LA1781, grown in different environments. Our results reveal that despite similar AS frequencies detected in the three genotypes, environment-responsive AS events are genotype-dependent, and the cultivated tomato Moneymaker has distinctive features of AS events in response to changing growth environments.

## 2. Results

### 2.1. The Transcriptome Dynamics of Inflorescences in Three Growth Environments

To better understand the developmental plasticity in cultivated tomato and its ancestral relatives, we counted flowers on eight consecutive inflorescences (I2-I9) starting from the 2nd inflorescence formed on individual plants grown in an open field (OF) and a plastic greenhouse (PG). Because the first inflorescences were developed before seedling transplantation and their growth could be impacted during the transplantation process, we did not count the flowers on the first inflorescences. Under both OF and PG growth conditions, the two accessions of the ancestral relative *S. pimpenillifolium*, LA1589 and LA1781, displayed considerable variations in flower number on individual inflorescences. The averaged numbers of flowers on I2 to I9 ranged from 19.2 to 26.2 and from 17.8 to 34.6 in PG-grown LA1589 and LA1781 plants with a coefficient of variation (CV) of 28.4% and 38.6%, respectively (Figure 1). Some LA1781 inflorescences did not even stop flower formation within the duration in which our experiments were conducted, becoming indeterminate (Figure 1C). In contrast, the numbers of flowers on the inflorescences I2 to I9 of the cultivated tomato LA2706 (Moneymaker) showed relatively smaller variations under both OF and PG conditions (under OF, CV = 21.9%; under PG, CV = 19.4%). Notably, the variations in flower number were more obvious on inflorescences I5 to I7 in PG-grown plants, whereas for the OF-grown plants, I3-I5 had relatively larger variations (Figure 1D–F).

Given the distinctive response of inflorescence development to growth conditions observed between cultivated tomato and its ancestral relatives, we analyzed the transcriptome dynamics of apical meristems (IM) and inflorescences (IF) in LA1589, LA1781 and LA2706 plants grown in PG and OF. Inflorescence samples (one replicate) from plants grown in phytotrons (PH) were included. Plants in PG and OF were grown in soil and those in PH were grown in peat. We generated more than 67.4 M paired-end reads from the 30 IM and IF libraries of the genotypes grown in PH, OF and PG, of which 93.11% concordant read pairs could be mapped on the tomato Heinz1706 reference genome (version ITAG4.0) (Appendix A). Using the uniquely mapped reads, we were able to assemble 34,460 genes from 105,638 transcripts including 1880 new transcription regions and 34,075 annotated genes, of which 17,776 were expressed (FPKM ≥ 1) in at least one of the tissues investigated. The numbers of expressed genes in IF were similar among the three genotypes grown in PH, PG and OF, ranging from 15,900 in the IM samples of PG-grown LA1589 plants to 16,275 in inflorescences of PH-grown LA2706 plants.

### 2.2. Alternative Splicing in Response to Growth Conditions

Alternative splicing (AS) is an important mechanism underlying the regulation of transcriptome complexity in response to environmental cues [11,30,32,52]. We first analyzed AS events predicted in the assembled transcripts. There were 122,471 AS events predicted in the transcripts assembled from all the sequence reads. Then, we surveyed the AS frequencies in each biological sample. The respective frequencies of AS detected in the three genotypes were 74.9%, 81.7% and 84.1% when the expression values (FPKM, fragments per kilobase per million reads) of the multi-exonic genes were set at 0.1, 0.5 and 1.0; the higher the cutoff of expression levels applied, the higher AS frequencies were obtained (Table 1). This indicates that the AS frequency in a particular sample is affected by gene expression level. For each genotype, the AS frequency was lower in the inflorescences of PH-grown plants than those of PG- or OF-grown plants. For example, with a cutoff of expression value of 1.0, 47.4%, 47.9% and 49.4% multi-exonic genes underwent AS, respectively, in the inflorescences of the PH-grown plants of LA1589, LA1781 and LA2706, whereas more than 57% genes underwent AS in the inflorescences from PG- and OF-grown plants of the three genotypes. Under PG growth conditions, similar AS frequencies were detected in the IM samples among the three genotypes and were also comparable to those detected in the IF samples. Overall, the AS frequencies in the inflorescences of the three genotypes were similar when plants were grown under the same conditions.

Though the AS frequencies in the IM and IF samples were similar among the three genotypes under the same growth condition, we tried to evaluate the potential difference in alternative splicing to the changes of growth conditions. To minimize the impact on AS by nutrient supply, we only compared AS events detected in the IF samples from PG and OF plants, which were grown in the soil of the same source and planted in a nearby field. We identified 298, 268 and 528 annotated multi-exonic genes showing differential AS (DAG) in the inflorescences of LA1589, LA1781 and LA2706, respectively (Figure 2A, Appendix A). Intriguingly, only five DAGs were shared by the three genotypes. Among them, *Solyc02g085420* encodes a member of the U1 small nuclear ribonucleoprotein family U1SNRNP that is likely involved in RNA splicing. *Solyc11g066830* encodes zinc finger transcription factor 68, and the remaining three genes *Solyc06g069020, Solyc01g091260* and *Solyc01g099010*, encode elongation factor, myeloid leukemia factor and GDSL esterase/lipase, respectively. In addition to the five DAGs shared by the three genotypes, there were also a few additional DAGs shared by any two of the three accessions; the cultivated tomato LA2706 shared 29 and 19 DAGs with two accessions of its ancestral relatives LA1589 and LA1781, respectively, and the latter two shared 12 DAGs (Appendix A). This suggests that the majority of these DAGs are genotype-specific.

Then, we performed a gene ontology (GO) analysis on the DAGs using PANTHER. We found that 20 GO terms (GO-Slim biological processes) were enriched in the DAGs of LA2706, whereas 15 and 6 GO terms were enriched in those of LA1781 and LA1589. GO terms enriched in the LA2706 DAGs included all GO terms enriched in the LA1589 and LA1781 DAGs except one that was only present in the LA1781 DAGs (Appendix A). However, if only considering the most specific terms (the narrowest child GO terms) enriched, no term was shared by the three genotypes (Figure 2B). The GO terms of the nitrogen compound metabolic process (GO:0006807), cellular metabolic process (GO:0044237) and organic substance metabolic process (GO:0071704) were enriched in the LA1589 DAGs, and only the RNA metabolic process (GO:0016070) was enriched in the LA1781 DAGs, whereas three categories including mRNA splicing via spliceosome (GO:0000398), the cellular nitrogen compound biosynthetic process (GO:0044271) and the cellular macromolecule metabolic process (GO:0044260) were enriched in those of LA2706. There were 17 splicing factor encoding genes in the category of mRNA splicing via spliceosome that was only enriched in the DAG list of LA2706, suggesting that AS regulation by environment was more prominent in LA2706. Thus, the GO enrichment analysis indicates that environment-responsive alternative splicing is genotype-specific and the cultivated tomato LA2706 has distinctive features of AS response, compared with its two ancestral relatives.

To assess the accuracy of the AS prediction, we analyzed the major AS events of the ten DAGs—the five shared by the three genotypes and five of the 17 RNA splicing factors in the list of LA2706 DAGs—by reverse transcription PCR (RT-PCR) in 12 RNA samples isolated from IM and IF of PH, PG and OF plants. Among them, the AS events could be validated by RT-PCR in eight DAGs (Figure 3). The AS events in *Solyc06g069020* and *Solyc05g007200* were not verified due to either unsuccessful amplification or small size difference between isoforms, respectively. The validation of major AS events by RT-PCR indicates that the AS prediction in this study is reliable.

### 2.3. Expression Features of Splice Variants of DAGs

Splice variants generated by AS events may encode new protein isoforms or lose protein coding potentials. To better understand the features of AS variants, we examined the expression levels and coding potentials of the individual AS variants of the above-mentioned ten DAGs. The 5 DAGs shared by the three genotypes had 2–8 AS variants (Figure 4). Four of them had at least two AS variants that encode different protein isoforms, whereas AS in the *Solyc11g066830* gene did not affect protein coding (Appendix A). Notably, the U1 small nuclear ribonucleoprotein encoding gene *Solyc02g085420* had six AS variants encoding five different protein isoforms and two having coding sequences identical to the annotated transcripts. This result suggests that AS regulates transcriptome complexity in at least two ways—increasing protein encoding compacity and mRNA abundance—in response to environmental changes. Since DAGs were identified based on the expression changes of individual isoforms, we found that for each gene only the expression of a few isoforms was responsive to environmental changes (Figure 4). For example, a different expression in the inflorescences was detected in five of the nine *Solyc02g085420* isoforms between PG and OF plants. Moreover, despite that the five DAGs were identified in all the three genotypes, the expression of several AS variants showed different responses to environmental changes. For example, one AS variant of *Solyc01g091260* had a higher expression in the inflorescences of LA2706 plants grown in PG, contrasting with its higher expression in those of LA1589 and LA1781 plants grown in OF.

Similarly, the five DAGs encoding splicing factors from the DAG list of LA2706 contains two to eight new splice variants, putatively encoding more than one new protein isoforms (Figure 5 and Appendix A). Despite that most splice variants were also detected in the inflorescences of LA1589 and LA1781 plants grown in PG and OF, the proportions of individual variants in total transcript abundance were only significantly affected in those of LA2706. Moreover, three of the five DAGs (*Solyc02g061840*, *Solyc03g026240* and *Solyc05g007200*) had new isoform-encoding variants showing differential expression in the LA2706 inflorescences of PG and OF plants, whereas those of *Solyc01g105140* and *Solyc02g066840* were barely expressed under the two growth conditions.

### 2.4. Differentially Expressed Genes and AS Variants Associated with the Plasticity of Inflorescence Development

We then checked the transcriptional response to growth conditions at the gene and isoform levels. Comparing PG and OF growth conditions, there were 871, 413 and 616 genes differentially expressed (DEGs) with a cutoff of transcript abundance FPKM ≥ 1 and the adjusted *p* value ≤ 0.05 in the IFs of LA1589, LA1781 and LA2706, respectively (Table 2). At isoform level, there were 1106, 599 and 1119 isoforms differentially expressed (DEIs) between PG and OF growth conditions in LA1589, LA1781 and LA2706, respectively (Table 2). Among these DEIs, 350 and 681 DEIs belonging to splice variants (differentially expressed variants, DEV) and annotated transcripts (differentially expressed annotated transcripts, DEAs) were detected in LA1589, respectively. Similarly, LA1781 had 225 DEVs and 342 DEAs, and LA2706 had 529 DEVs and 505 DEAs. Though there were 97 DEGs and 80 DEIs shared by the three genotypes, most DEGs and DEIs were genotype-specific, only detected in one particular genotype, except about two-third of the DEGs from LA1781 were detected in either LA1589 or LA2706 (Figure 6A,B; Appendix A). These results suggest that growth conditions impact the transcription of different sets of genes in the three genotypes.

We then performed GO analysis on these DEGs and genes having DEIs. Similar to the results from GO analysis on DAGs, the GO terms enriched in the DEGs and DEIs were very different among the three genotypes when only the narrowest child GO terms were considered (Figure 6C,D). For example, genes involved in the pigment biosynthetic process (GO:0046148), photosynthesis (GO:0015979), post-embryonic development (GO:0009791), the generation of precursor metabolites and energy (GO:0006091) and protein folding (GO:0006457) were over-represented in the DEGs of LA1589. Genes involved in the circadian rhythm (GO:0007623), cellular response to heat (GO:0034620), cellular response to unfold protein (GO:0034620) and chaperone cofactor-dependent protein refolding (GO:0051085) were enriched in the DEGs of LA1781. For the DEGs detected in LA2706, the enriched GO terms were the circadian rhythm, the response to heat (GO:0009408), the response to osmotic stress (GO:0006970), chaperone-mediated protein folding (GO:0061077), the response to oxygen-containing compounds (GO:1901700) and the lipid metabolic process (GO:0006629). However, when all enriched terms were included, there were several categories shared by the three genotypes, i.e., the response to abiotic stimulus (GO:009628), protein folding (GO:0006457) and the response to chemicals (GO:0042221) in the DEGs, and the response to stimuli (GO:0050896) in the DEIs of the three genotypes (Appendix A). Interestingly, LA2706 shared more categories of enriched GO terms with LA1781 than with LA1589.

Further checking the expression of the 97 DEGs shared by the three genotypes, we found that most of them had similar expression patterns—either up-regulated or down-regulated in the IFs of PG plants; only eight DEGs showed different patterns between LA2706 and its ancestral relatives, LA1589 and LA1781 (Figure 6E). The distinctive response of the eight genes may be associated with the difference in the plasticity of inflorescence development observed between cultivated tomato and its ancestral relatives. Seven of the eight DEGs showing different expression patterns were involved in heat response. For example, there were three DEGs encoding heat shock proteins (*Solyc11g066100, Solyc06g076020, Solyc06g036290*) and two encoding heat shock transcription factors (*Solyc09g065660/HsfA7*; *Solyc08g080540/HsfB-2b*). Expression of these genes was up-regulated in the IFs of LA1589 and LA1781 plants grown in OF but down-regulated in LA2706. The different response of these heat-induced genes to growth conditions in cultivated tomato and its ancestor relatives indicates that the transcription regulation of heat response may be targeted by domestication.

## 3. Discussion

Domestication and breeding improvements have profound impacts on reproductive development in tomatoes. Modern tomatoes, especially large-fruited varieties (i.e., Moneymaker/LA2706), produce relatively stable numbers of flowers/fruits on each inflorescence, whereas their ancestral relatives *S. pimpinellifolium* (i.e., LA1589 and LA1781) exhibit strong plasticity of inflorescence development under different growth environments [36,37] (Figure 1). This suggests that there may be considerable differences in environment-responsive gene expression associated with the plasticity of inflorescence development between cultivated tomatoes and their ancestral relatives. Previous studies have identified several genes regulating inflorescence complexity and flower number in tomatoes [38,39,40,41,42,43,44,45]. However, none of these characterized genes showed significant differences in expression patterns in response to changing environments among the three genotypes, suggesting that these genes are unlikely responsible for the inflorescence plasticity observed in the three genotypes. In this study, we identified two genes encoding heat stress transcription factors HsfA7 and HsfB2b and three genes encoding heat shock proteins whose expressions showed opposite responses between Moneymaker and *S. pimpinellifolium* accessions LA1589 and LA1871 (Figure 6). Cultivated tomatoes have large apical bud structure which may affect meristem temperature, which has been thought to be drastically impacted by apical bud structure and function [53]. Since *HsfA7* is a capacitor of the central regulator of *HsfA1* during the early heat stress response [50], its distinctive expression response may be caused by the putative difference in meristem temperature between Moneymaker and its ancestral relatives. Notably, plant temperature is one of the key modulators controlling plant development and heat stress transcription factors (i.e., *HSFA2* in Arabidopsis) have been shown to regulate heat stress-related SAM development [54,55]. Given the plastic greenhouse had relative higher temperature than the open field, it is possible that the distinctive environment-responses of *HsfA7* and *HsfB2b* expression are associated with the difference in the plasticity of inflorescence development between cultivated tomatoes and *S. pimpinellifolium* accessions. This possibility will be explored through a functional analysis of the two *Hsf* genes and/or the three *HSPs*.

AS plays a crucial role in the regulation of plant fitness under abiotic stresses [51]. Our previous transcriptome analysis of seedlings has also demonstrated that AS plays an important role in regulation of environment-responsive gene expression in seedlings [36]. In this study, very similar AS frequencies were detected in the inflorescences of the three genotypes when grown under the same conditions, but only five genes showing differential splicing between OF and PG conditions were found in all the three genotypes (Figure 2). This suggests that the AS response of the inflorescence to changing environments is genotype-dependent, which is in agreement with our previous AS analysis conducted in seedlings [36]. Interestingly, many more DAGs were detected in Moneymaker than in LA1589 and LA1781. Notably, there were 17 splicing factors including five encoding RRM domain-containing proteins that showed differential splicing in Moneymaker. Stresses often induce AS in some splicing factors that change the splicing patterns of their targets, which likely contribute to plant adaptation to stresses [56,57]. Because the targets of these splicing factors are still unknown, it is difficult to predict which splicing factors are involved in the regulation of the plasticity of inflorescence development. However, more splicing factors showing differential splicing in Moneymaker likely explain why more DEVs were detected in this genotype (Table 2). Therefore, the overrepresentation of splicing factors and the higher number of DAGs detected in Moneymaker inflorescences indicate that domestication and breeding improvement have likely changed the tomato’s AS response to its environments.

We detected not only more DAGs but also almost double the number of DEVs in Moneymaker inflorescences when compared with the two accessions of its ancestral relative. The distinctive response of splice variants is likely associated with the difference in the plasticity of inflorescence development between cultivated tomato and its ancestral relative *S. pimpinellifolium*. Particularly, one splice variant from *Solyc02g089200/TM29* was only differentially expressed in the inflorescences of Moneymaker (Appendix A). *TM29* has been shown to regulate the maintenance of floral meristem identity [58]. MADS-box genes are hypothesized to be key transcriptional regulators of developmental plasticity in response to seasonal changes in environmental conditions [59]. The expression of some tomato MADS genes is regulated by temperature [60]. Thus, the response of some MADS transcription factors at the gene and/or isoform levels may be associated with the plasticity of inflorescence development. In addition, one splice variant of *HsfA2* was only differentially expressed in Moneymaker, which is in agreement with previous report that natural variations in *HsfA2* splicing are associated with the changes in thermotolerance during tomato domestication [51].

A limitation of this study is that the environment-responsive AS events were only investigated in one tomato cultivar and two accessions of a wild tomato species. Further detailed AS analysis in a larger number of representative tomato accessions together with genetic analysis may help pinpoint the candidate genes controlling the plasticity of inflorescence development and identify AS events associated with the variations between cultivated tomatoes and the ancestral relatives. Nevertheless, our findings shed new light on the regulation of gene expression mediated by alternative splicing in response to changing growth conditions.

## 4. Materials and Methods

### 4.1. Plant Growth Conditions

The wild tomato species accessions *S. pimpinellifolium* LA1589 and LA1781 and cultivated tomato cv. Moneymaker (LA2706) were obtained from the Tomato Genetics Resource Center at the University of California, Davis, CA, USA. The three accessions were grown in three different environments: a well-controlled phytotron, an open field and a plastic greenhouse—a type of solar greenhouse used for vegetable production. When grown in a phytotron, plants were potted in blonde peat (Pindstrup Mosebrug A/S, Ryomgaard, Denmark) and maintained at 18–25 °C with a humidity of 70%–80% under daily illumination of 150 mE/m^2^/s for 16 h. Plants grown in an open field (OF) and a plastic greenhouse (PG) were under natural solar radiation. The plastic greenhouse and open field used for growing plants were located in Songjiang Wushe, Shanghai. The temperature and humidity of OF and PG were monitored daily during plant growth season (April to June, 2016, Shanghai). PG temperature ranged from 15.6–29.1 °C with a median of 22.2 °C, while OF temperature ranged from 9.8–35.8 °C with a median of 21.3 °C. The relative humidity ranged from 32.0–90.4% (PG) and 33.7–83.2% (OF), respectively.

### 4.2. RNA Isolation and Library Construction

Total RNA was isolated from the inflorescence meristems (IM) of 15-day-old seedlings and the developing inflorescence (IF) of 45-day-old plants using Trizol reagent (Thermo Fisher Scientific, Waltham, MA, USA) based on the methods described previously [61]. For each sample, tissues from more than 30 plants were pooled as a biological replicate. With the exception of one replicate for IM from phytotron-grown seedlings (PH), three replicates were sampled for plants grown in the open field (OF) and the plastic greenhouse (PG). Library construction and sequencing have been described previously (Wang et al., 2017). Essentially, pair-end RNA-seq libraries were generated from 1 μg total RNA using NEBNext Ultra Directional RNA Library Prep Kit for Illumina (E7420L, New England Biolabs, Inc., Ipswich, MA, USA) and were sequenced on an Illumina Hiseq2500 system using Hiseq SBS Kit V3 (Illumina Inc., San Diego, CA, USA).

### 4.3. Data Processing

The quality of raw reads was first checked by fastqc (version v0.11.9, http://www.bioinformatics.babraham.ac.uk/projects/fastqc/), following the removal of low-quality reads and adaptor sequences by Trimmomatic (version 0.39) [62]. The processed clean reads were then mapped to the tomato reference genome (version ITAG4.0; Sol Genomics Network (SGN): solgenomics.net/) using Hisat2 (version 2.2.1) [63] with the following parameters: hisat2 -p 4 –dta-cufflinks -t –qc-filter –new-summary –rna-strandness RF -x. The resulting BAM files of read mappings were sorted and indexed by samtools (version 1.13) [64]. Then, transcripts were assembled by cufflinks (version v2.2.1) [65] using parameters: cufflinks -p 2 -u -g ~/genome/ITAG4.0_gene_models.gtf -b ~/genome/S_lycopersicum_chromosomes.4.00.fa --library-type fr-firststrand.

The identification of alternative splicing events, differentially expressed genes and isoforms (DEGs and DEIs), and differential splicing genes (DAG) were conducted based on our previously described procedures [21]. Briefly, DEGs, DEIs and DAGs were identified by the cuffdiff program in the cufflinks package using the following parameters: cuffdiff -o cfdiff_result/ -b ~/genome/S_lycopersicum_chromosomes.4.00.fa -u -p 40 --library-type fr-firststrand. DEGs and DEIs with adjusted *p*-values smaller than 0.05 selected by Cufflink were further filtered by the following criteria: expression values (FPKM) ≥ 1 and fold changes ≥ 2.

Enriched gene ontology (GO) terms in the gene list of DEGs, DEIs and DAGs were identified using the online tool PANTHER (Protein ANalysis THrough Evolutionary Relationships) Classification System (http://pantherdb.org/) [66]. For the GO analysis of DEIs, their corresponding gene accessions were used despite that some DEIs may encode different proteins.

### 4.4. RT-PCR Verification of AS Events

Gene-specific primers were designed to detect major AS events having size differences of more than 30 bp (primer information can be found in Appendix A). One microgram of total RNA for each sample isolated from inflorescence meristems and inflorescences using Trizol reagent (Thermo Fisher Scientific, Waltham, MA, USA) as mentioned above were subjected to cDNA synthesis in 20 μl reaction volume using MLV reverse transcriptase (New England Biolabs, Inc., Ipswich, MA, USA) after DNA removal by RNase-free DNase (New England Biolabs, Inc., Ipswich, MA, USA). Ten percent of the synthesized cDNA mixtures were used as templates for the detection of AS events.

## Figures and Tables

**Figure 1 ijms-23-11585-f001:**
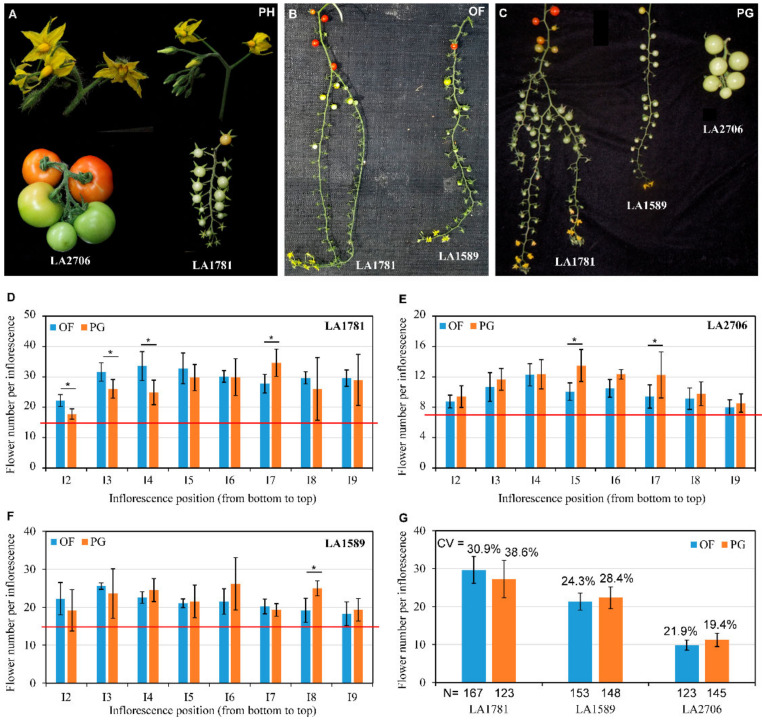
The plasticity of inflorescence development in three genotypes. (**A**), Representative inflorescences bearing flowers or fruits of LA1781 and LA2706 grown in phytotron (PH). (**B**), Inflorescences of LA1781 and LA1589 grown in open field (OF). (**C**), Inflorescences of LA1781, LA1589 and LA2706 grown in plastic greenhouse (PG). (**D**), Flower numbers per inflorescence on LA1781 plants grown in open field (OF) and plastic greenhouse (PG). (**E**), Flower numbers per inflorescence on LA2706 plants grown in open field (OF) and plastic greenhouse (PG). (**F**), Flower numbers per inflorescence on LA1589 plants grown in open field (OF) and plastic greenhouse (PG). (**G**), Averaged flower numbers on the 2nd to 9th inflorescences from plants grown in open field (OF) and plastic greenhouse (PG). Flowers were counted from the 2nd to 9th inflorescence on each plant. The data were collected from plants grown in four different blocks in an open field and four units of a plastic greenhouse, the number of plants investigated are indicated in (**G**). The red lines in (**D**–**E**) indicate the means of flower number per inflorescence from plants grown in phytotron (PH). CV, coefficient of variations. Error bars in (**D**–**F**) represent standard deviations of means. *, *p*-value ≤ 0.05 based on Student’s *t*-test.

**Figure 2 ijms-23-11585-f002:**
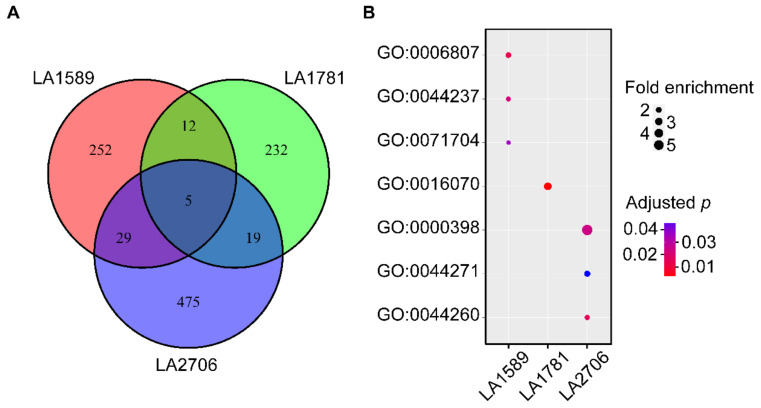
AS events in response to growth conditions**.** (**A**), A Venn diagram showing DAGs identified in the inflorescences of LA1589, LA1781 and LA2706 (Moneymaker) in response to changing growth conditions. (**B**), Gene ontology analysis of DAGs. DAGs were identified by comparison of AS events between open field and plastic greenhouse. GO:0006807, nitrogen compound metabolic process; GO:0044237, cellular metabolic process; GO:0071704, organic substance metabolic process; GO:0016070, RNA metabolic process; GO:0000398, mRNA splicing via spliceosome; GO:0044271, cellular nitrogen compound biosynthetic process, GO:0044260, cellular macromolecule metabolic process.

**Figure 3 ijms-23-11585-f003:**
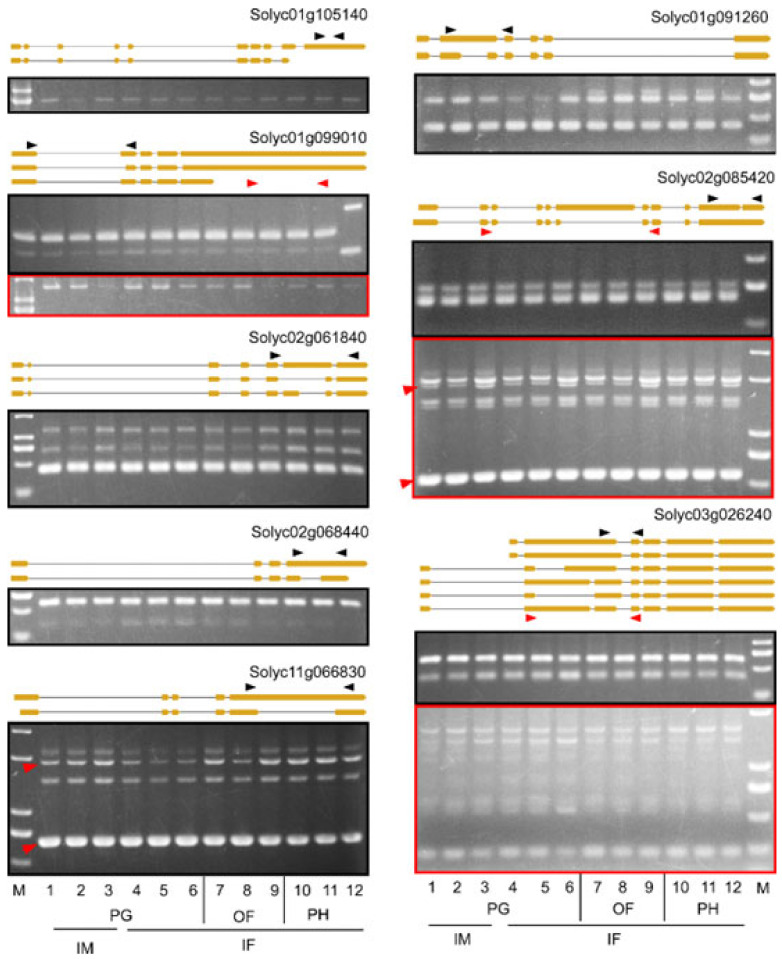
RT-PCR verification of several major splice variants identified by RNA-seq. AS events predicted in ten DAGs were selected for RT-PCR verification. The figure shows splice variants successfully verified in eight DAGs. The positions of primers to specifically amplify regions having AS events were indicated by arrowheads either above or beneath the gene models. Gel images boxed in black and red showed RT-PCR amplifications using primers indicated by the same colors. Sample loading sequence: LA1589 (1, 4, 7, 10); LA1781 (2, 5, 8, 11); LA2706 (3, 6, 9, 12). M, DNA size marker.

**Figure 4 ijms-23-11585-f004:**
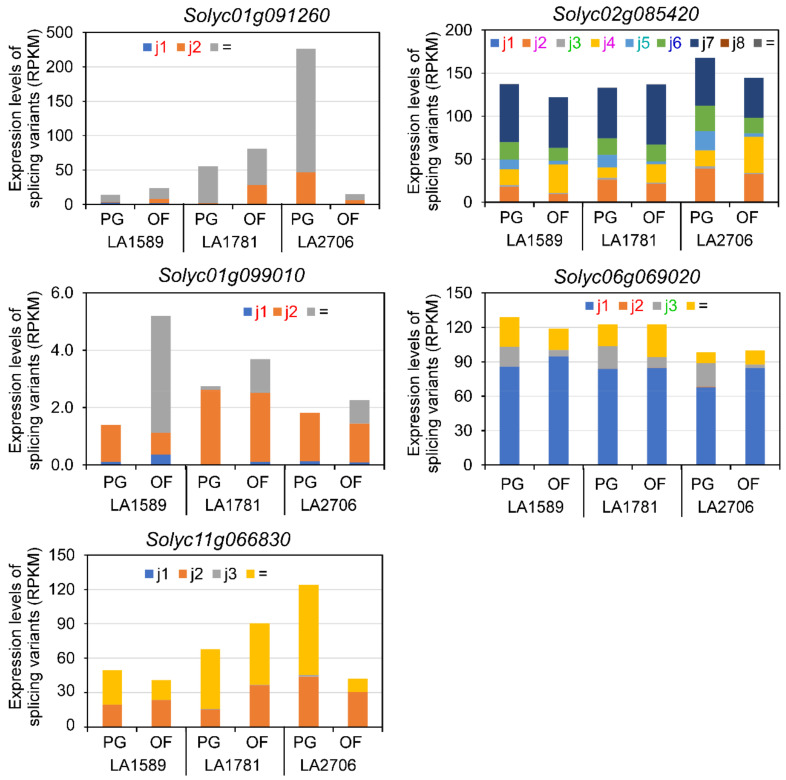
Expression levels of splice variants of the five DAGs found in LA1589, LA1781 and LA2706. These histograms show isoform abundance measured by RNA-seq. “=” represents isoform that is the same as annotated. “j” represents novel splice variant.

**Figure 5 ijms-23-11585-f005:**
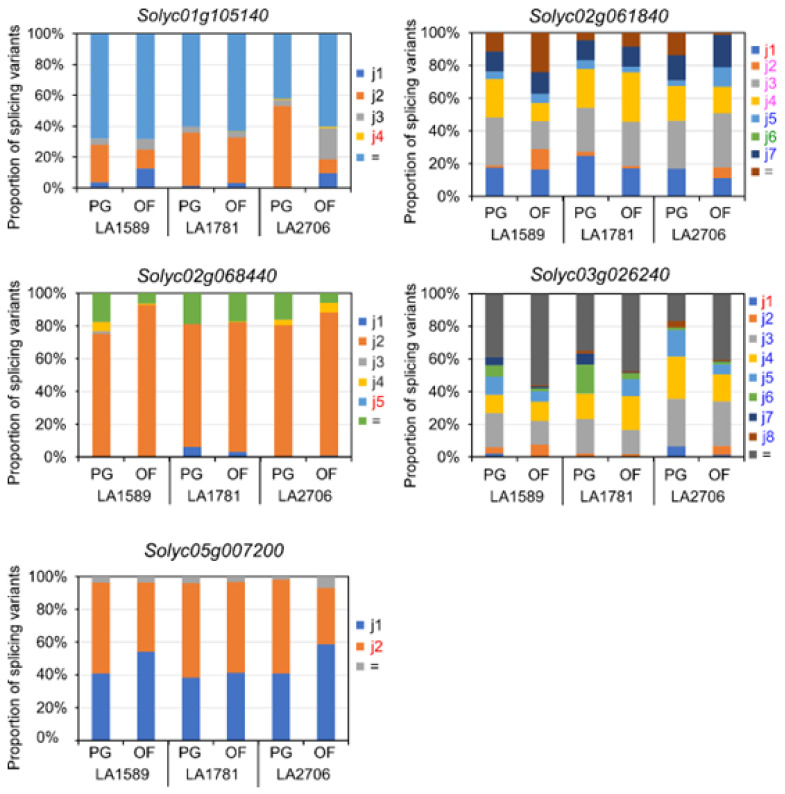
Expression levels of splice variants of five representative DAGs encoding splicing factors. The histograms show expression proportion of each splice variant. “=” represents an isoform the same as annotated. “j” represents a novel splice variant.

**Figure 6 ijms-23-11585-f006:**
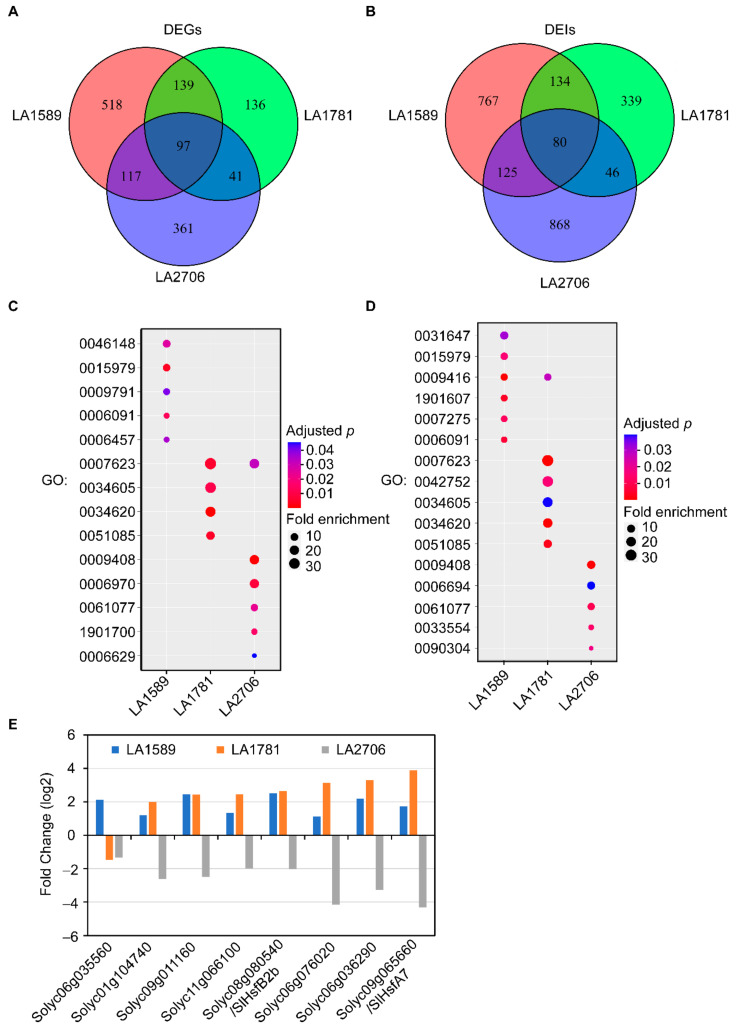
Comparison of gene expression in response to growth conditions in LA1589, LA1718 and LA2706. (**A**), A Venn diagram showing DEGs identified in the inflorescences of LA1589, LA1781 and LA2706 (Moneymaker) in response to changing growth conditions. (**B**), A Van diagram showing DEIs identified in the inflorescences of LA1589, LA1781 and LA2706 (Moneymaker) in response to changing growth conditions. (**C**), Gene ontology analysis of DEGs. (**D**), Gene ontology analysis of DEIs. DEGs and DEIs were identified by comparison of expression values between open field and plastic greenhouse. (**E**), Distinctive response of gene expression between LA2706 and its ancestral relatives. Following are the GO term names for the GO IDs present in (**C**,**D**): pigment biosynthetic process (GO:0046148), photosynthesis (GO:0015979), post-embryonic development (GO:0009791), generation of precursor metabolites and energy (GO:0006091), protein folding (GO:0006457), circadian rhythm (GO:0007623), cellular response to heat (GO:0034605), cellular response to unfolded protein (GO:0034620), chaperone cofactor-dependent protein refolding (GO:0051085), response to heat (GO:0009408), response to osmotic stress (GO:0006970), chaperone-mediated protein folding (GO:0061077), response to oxygen-containing compound (GO:1901700), lipid metabolic process (GO:0006629), regulation of protein stability (GO:0031647), response to light stimulus (GO:0009416), alpha-amino acid biosynthetic process (GO:1901607), multicellular organism development (GO:0007275), regulation of circadian rhythm (GO:0042752), steroid biosynthetic process (GO:0006694), cellular response to stress (GO:0033554) and nucleic acid metabolic process (GO:0090304). Only the narrowest GO terms were present in (**C**,**D**); complete lists of enriched GO terms can be found in Appendix A.

**Table 1 ijms-23-11585-t001:** The numbers of multiexon genes underwent AS in IM and IF from OF, PG and PH grown plants.

Gene Expression Levels	FPKM ≥ 0.1			FPKM ≥ 0.5			FPKM ≥ 1.0		
	Number of expressed multiexon genes	Number of genes with AS	AS Frequency	Number of expressed multiexon genes	Number of genes with AS	AS Frequency	Number of expressed multiexon genes	Number of genes with AS	AS Frequency
LA1589_IF_PG	15415	8005	51.93%	14240	7984	56.07%	13629	7919	58.10%
LA1589_IF_OF	15397	7955	51.67%	14238	7931	55.70%	13625	7860	57.69%
LA1589_IF_PH	15309	6552	42.80%	14286	6545	45.81%	13726	6501	47.36%
LA1589_IM_PG	15331	7946	51.83%	14179	7938	55.98%	13565	7895	58.20%
LA1781_IF_PG	15356	8167	53.18%	14285	8146	57.02%	13691	8066	58.91%
LA1781_IF_OF	15439	8361	54.16%	14320	8289	57.88%	13703	8143	59.42%
LA1781_IF_PH	15513	6674	43.02%	14391	6657	46.26%	13813	6613	47.88%
LA1781_IM_PG	15260	7866	51.55%	14158	7853	55.47%	13567	7800	57.49%
LA2706_IF_PG	15369	8097	52.68%	14201	8081	56.90%	13616	8019	58.89%
LA2706_IF_OF	15412	8299	53.85%	14265	8241	57.77%	13682	8101	59.21%
LA2706_IF_PH	15499	6913	44.60%	14415	6895	47.83%	13834	6836	49.41%
LA2706_IM_PG	15419	7746	50.24%	14297	7734	54.10%	13693	7670	56.01%
Total	17100	12811	74.92%	15570	12720	81.70%	14923	12551	84.11%
PG, plastic greenhouse; OF, open field; PH, phytotron									

**Table 2 ijms-23-11585-t002:** Summary of differentially expressed splice variants, isoforms and genes in the inflorescences under PG and OF conditions.

	LA1589IF	LA1781IF	LA2706IF
Number of expressed genes (FPKM ≥ 1)	17947	17946	17984
Number of DEGs	871	413	616
Number of DEIs	1106	599	1119
Number of DEVs	350	225	529
Number of DEAs	681	342	505

DEGs, differentially expressed genes; DEI, differentially expressed isoform/transcript; DEV, differentially expressed splicing variants; DEA, differentially expressed annotated transcripts.

## Data Availability

Raw sequence reads obtained in this study were deposited in the Genome Sequence Archive (GSA, https://ngdc.cncb.ac.cn/gsa**/**) (accession: CRA007845). All data will be available upon request to corresponding author (H.X.).

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
