# Peer review of "Comparative Analysis of Environment-Responsive Alternative Splicing in the Inflorescences of Cultivated and Wild Tomato Species"

_ijms, 2022, doi:10.3390/ijms231911585_

Round 1
Reviewer 1 Report
All comments, queries and needed edits are given in the pdf version.

Author Response
Thank you for your comments and suggestions. Please see our revision and response file (pdf).

Reviewer 2 Report
There are some concerns with the study, which I have added pointwise:
1) the Third line of the result section needs a spelling check.
2) Please add a significance test on Fig 1(A), (B), (C) graphs. Maybe an ANOVA test.
3) Please add representative images of inflorescences from LA1781, LA1589, and LA2706 in different conditions.
4) Quality of the exported images is not good. For e.g, Fig 6(C), it is impossible to read. There are multiple examples like this. Please revisit and export as TIF or PNG.
5)Authors show a difference in inflorescence development in the modern variety and domesticated tomato. So, that means there should be relation between the expression of reproductive genes in both female and male gametophytes-related genes. Is there any gene differentially expressed and is related to fertility? There should also be a paragraph about male and female meiosis related to fertility.
6)If there is a change in inflorescence number, then what is the phenotypic difference in length of tomato plants? Please provide images.
Author Response
Thank you for your comments and suggestions.
1) the Third line of the result section needs a spelling check.
A: corrected.
2) Please add a significance test on Fig 1(A), (B), (C) graphs. Maybe an ANOVA test.
A: Statistical analysis (T.Test) was conducted. We also calculated Coefficient of Variances (CV) for comparing the degree of variation in flower number per inflorescence between OF and PG plants. Figure 1 has been revised.
3) Please add representative images of inflorescences from LA1781, LA1589, and LA2706 in different conditions.
A: we have shown these images in our previous publication (Wang et al. Response of gene expression and alternative splicing to distinct growth environments in tomato. Int J Mol Sci, 2017. 18(3)).
4) Quality of the exported images is not good. For e.g, Fig 6(C), it is impossible to read. There are multiple examples like this. Please revisit and export as TIF or PNG.
A: high resolution images (tiff format) have been uploaded separately.
5)Authors show a difference in inflorescence development in the modern variety and domesticated tomato. So, that means there should be relation between the expression of reproductive genes in both female and male gametophytes-related genes. Is there any gene differentially expressed and is related to fertility? There should also be a paragraph about male and female meiosis related to fertility.
A: That is a good question. We in this study did not intend to address this question. We mainly focused on the flower number per inflorescence as an indication of inflorescence development. Our goal is to investigate whether transcriptome dynamics is associated with the difference in the plasticity of inflorescence development observed between cultivated tomatoes and the ancestral relatives S. pimpenillifolium.
6)If there is a change in inflorescence number, then what is the phenotypic difference in length of tomato plants? Please provide images.
A: Since the three genotypes are indeterminate type, we believe plant height is not informative.
Reviewer 3 Report
The manuscript “Comparative analysis of alternative splicing associated with phenotypic stability in cultivated and wild tomato species” reveals characteristic features of alternative splicing (AS) events in the inflorescences between cultivated tomato and its ancestral relatives, and also show the impact of AS events in response to changing environments by domestication and breeding improvement. The article is well-written and has limitations in experimental design and novelty. The authors themselves acknowledged that the study is performed in very few genotypes with little or no replications, and since the AS response is genotype-dependent as suggested by the authors, it becomes even more critical to include a larger set of genotypes. Similarly, the authors and other researchers have published several kinds of these studies, hence there is a lack of novelty. Therefore, it is recommended that the authors perform the study with more genotypes, and Although representation is very good, some figures are not readable, and hence need to be improved.
Author Response
The manuscript “Comparative analysis of alternative splicing associated with phenotypic stability in cultivated and wild tomato species” reveals characteristic features of alternative splicing (AS) events in the inflorescences between cultivated tomato and its ancestral relatives, and also show the impact of AS events in response to changing environments by domestication and breeding improvement. The article is well-written and has limitations in experimental design and novelty. The authors themselves acknowledged that the study is performed in very few genotypes with little or no replications, and since the AS response is genotype-dependent as suggested by the authors, it becomes even more critical to include a larger set of genotypes. Similarly, the authors and other researchers have published several kinds of these studies, hence there is a lack of novelty. Therefore, it is recommended that the authors perform the study with more genotypes, and Although representation is very good, some figures are not readable, and hence need to be improved.
A: Thank you for your comments and suggestion.
As we indicated in the manuscript, we agree that RNA-seq from more genotypes will need to pinpoint the candidate genes controlling the plasticity of inflorescence development. That experiments need more than one year. Here, we intend to identify AS events associated with the variations in inflorescence development between cultivated tomatoes and the ancestral relatives. Our and other previous papers only reported AS analysis on seedlings, not reproductive tissues and did not touch the association between the plasticity of inflorescence development and environments. Also, this study provides new evidences to show that AS response to changing environments was shaped by domestication and breeding in tomato.
Figures of high resolution have been uploaded.
Round 2
Reviewer 2 Report
I have no further comments.
Author Response
Thank you
Reviewer 3 Report
As the authors have pointed out, that the present study provides new evidences to show that AS response to changing environments was shaped by domestication and breeding in tomato; however, since the response is genotype dependent, the inferences drawn would provide and confirm the results only with a larger set of genotypes.
Author Response
Thank you for the comments. We essentially agree that data collected from more samples will be more informative. Nevertheless, we think our RNA-seq analysis on the three genotype is informative. Also, collecting such data will take more than one years.